# Torsional Vibration Tests of Extruded Polystyrene with Improved Accuracy in Determining the Shear Modulus

**DOI:** 10.3390/polym14061148

**Published:** 2022-03-13

**Authors:** Hiroshi Yoshihara, Momoka Wakahara, Masahiro Yoshinobu, Makoto Maruta

**Affiliations:** 1Faculty of Science and Engineering, Shimane University, Nishikawazu-cho 1060, Matsue 690-8504, Japan; waka0410momoka@yahoo.co.jp (M.W.); yosinobu@riko.shimane-u.ac.jp (M.Y.); 2Faculty of Science and Technology, Shizuoka Institute of Science and Technology, Toyosawa 2200-2, Fukuroi 437-8555, Japan; maruta.makoto@sist.ac.jp

**Keywords:** extruded polystyrene, shear modulus, torsional vibration test, flexural vibration test, anisotropy

## Abstract

Recently, extruded polystyrene (XPS) foam has been used as a component of construction materials; therefore, it is important to characterize its mechanical properties, including shear modulus. Despite the importance, it is often difficult to determine the shear modulus accurately by using many of the conventional methods; therefore, it is desirable to establish another method to measure the shear modulus with a high accuracy. Among various methods, torsional vibration test is advantageous because it can be performed easily under the pure shear stress condition in the test sample and both the in-plane and out-of-plane shear moduli can be obtained. However, it is difficult to find any examples performing the torsional vibration tests. In this study, the in-plane and out-of-plane shear moduli of XPS were determined through torsional vibration tests using samples of various widths. In addition, the shear moduli were also determined through flexural vibration tests and compared with those obtained from the torsional vibration tests. In the torsional vibration tests, the anisotropy in these shear moduli became an obstacle, and the in-plane shear modulus determined using a single sample was often dependent on the width/thickness ratio of the sample. In this condition, the coefficient of variation of the in-plane shear modulus value was often close to 10%. However, when using data obtained from the samples with various width/thickness ratios, both the in-plane and out-of-plane shear moduli could be obtained while reducing the abovementioned dependence. Additionally, the coefficients of variation were restricted to those below 2% and 7% for the in-plane and out-of-plane shear moduli, respectively, and these values were obviously lower than those obtained from the flexural vibration tests (approximately 20%). In the proposed method, both the in-plane and out-of-plane shear moduli can be obtained accurately without using any numerical analyses, which are often required in the standardized methods to improve the accuracy. Thus, for accurate measurement of both types of shear moduli of XPS, we recommend performing torsional vibration tests using a range of samples of various width/thickness ratios.

## 1. Introduction

Extruded polystyrene (XPS) foam has been conventionally used as a heat insulating material because of its high thermal insulation performance [1,2,3,4,5]. Additionally, the weightlessness of XPS is expected to allow attenuation of seismic forces; therefore, XPS can be used as a component of numerous construction materials, such as sandwich panels [6,7,8,9], flooring materials [10,11,12], and geofoams [13,14,15]. To ensure XPS is appropriately used in a structure, it is necessary to characterize its mechanical properties, including the shear properties.

There are several examples examining the shear modulus of foam materials, such as XPS and expanded polystyrene (EPS). Table 1 lists the representative values of shear modulus obtained from XPS and EPS under various methods. It is often difficult to obtain the relevant shear properties, and this difficulty is enhanced when the shear test is conducted under static loading conditions. For example, the in-plane shear test methods of sandwich core materials and thermal insulating products standardized in ASTM C273/C273M-11 [16] and EN 12090 [17] can be applied to measure the shear properties of foam materials. In these methods, a shearing force is directly applied to the sample via the fixtures fixed to the sample. As listed in Table 1, these methods are adopted by Kilar [14], Yoshihara et al. [18], Taskin et al. [19], Gnip et al. [20], and Vėjelis el al. [21]. However, it is often difficult to accurately determine the shear properties by using this method because of the concentration of stress between the test sample and fixture bonded onto the sample to apply the shearing force. The shear deformation is more enhanced in the region near the fixture because of the stress concentration, and the shear modulus tends to be measured as small because of the large deformation. In ISO 15310, the square-plate twist method of fiber-reinforced plastic composites is standardized, and this method was applied to measure the in-plane shear modulus [22]. Although the abovementioned stress concentration can be reduced in this method, it is difficult to obtain the in-plane shear modulus directly from the test because of the effect of out-of-plane shear modulus, and finite element (FE) calculations were required to improve the accuracy. In addition to these standardized methods, asymmetric four-point bending (AFPB) tests were conducted [23]. As shown in Table 1, the shear modulus value obtained from the AFPB test is higher than those obtained from the EN 12090, ASTM C273, and ISO 15310 methods. However, in the AFPB test, a strain gage is bonded onto the test sample, and strain gage calibration is required independently of the asymmetric four-point bending test. Although optical methods such as digital image correlation (DIC) and the virtual fields method (VFM) are effective for obtaining the strain distribution in a porous material [24,25,26]. However, these optical methods require the special equipment, and, therefore, they are not always cost effective.

The abovementioned obstacles of static tests are due to difficulty in the relevant determination of the shear stress–strain relation. However, when measuring the shear modulus alone, several dynamic methods, including vibration techniques, are effective. The resonant column method adopted by Athanasopoulous et al. [27] and Ossa et al. [28] and the ultrasonic method by Abd-El-Mottaleb et al. [29] may be promising to obtain the shear modulus. However, in these methods, anisotropy in the test material was not taken into account; therefore, further research should be conducted to obtain the shear modulus accurately with considering the effect of anisotropy. In previous studies, flexural vibration (FV) tests of beam-shaped samples [18] and torsional vibration tests of square-plate samples [30] were performed, and several promising results were obtained. However, the configuration of the sample affected the accurate measurement of shear modulus. Therefore, similar to the ISO 15310 method, FE calculations were required to improve the accuracy.

Considering the simplicity and practicality, it is desirable to determine the shear modulus accurately from the testing data alone without performing any FE calculations. Additionally, the method for measuring the shear modulus of XPS has not been determined in major standards, including ASTM, EN, ISO, and JIS. When considering the extension of the use of XPS as construction materials, it is necessary to establish the measurement method of the shear modulus for future standardization. To measure the shear modulus, torsional vibration test is more advantageous than other methods because it can be performed easily under the pure shear stress condition in the test sample. Therefore, when the effect of sample configuration can be reduced, the shear modulus can be measured with a high accuracy, and it is promising that the torsional vibration test becomes a candidate for measuring the shear modulus of XPS in the major standards. However, it is difficult to find the examples applying the torsional vibration tests to measure the shear modulus of XPS.

In this study, both the in-plane and out-of-plane shear moduli of the samples cut from an XPS panel were measured via the torsional vibration tests, as well as the flexural vibration tests. In the torsional vibration tests, data obtained from a single sample and multiple samples with various widths were used, and an optimal method to obtain these shear moduli was developed.

## 2. Materials and Methods

### 2.1. Materials

An XPS panel (STYROACE-II, Dupont Styro Corporation, Tokyo, Japan) was used for the tests. The initial dimensions of the panel were 1820, 910, and 25 mm in the length, width, and thickness directions, respectively. The length, width, and thickness directions of the panel were determined as L, T, and Z axes, respectively. A heat wire was used to cut the test samples. The density of the sample was 29.4 ± 0.5 kg/m^3^.

### 2.2. Torsional Vibration Tests

Torsional vibration tests were performed to obtain the in-plane and out-of-plane shear moduli. The initial length of the sample was 300 mm, whereas the initial width was 150 mm. The samples whose length direction coincided with the L and T axes were defined as L- and T-type samples, respectively. Five samples were used per testing condition. After performing the torsional vibration tests, the widths of the samples were decreased, and torsional vibration tests were consecutively performed. The width was decreased from 150 to 50 mm in intervals of 25 mm. In the torsion test of anisotropic material, the effects of the in-plane and out-of-plane shear moduli vary according to the aspect ratio *B/H* [31]. Therefore, it is expected to obtain both shear moduli, which can be measured when using the data obtained from multiple torsion tests while varying the *B/H* value.

Figure 1a,b represents the setup and diagram of the torsional vibration test, respectively. The sample was supported by using four prismatic wood blocks fixed on plywood at the midpoints of the length and width of the sample, which correspond to the nodes of the 1st torsional vibration mode. Torsional vibration was generated along the thickness direction at one corner of the sample, and it was detected by a microphone (UC-53AH, Rion Co., Tokyo, Japan) with a pre-amplifier (NH-22, Rion Co., Tokyo, Japan) at the diagonal corner to the generated corner. A fast Fourier transform (FFT) analyzer (SA-78, Rion Co., Tokyo, Japan) was used to obtain the fundamental frequency of the torsional vibration mode. The *x*, *y*, and *z* directions coincided with the length, width, and thickness direction of the sample, respectively. The in-plane shear modulus, *G_xy_*, is rigorously derived as follows [31]:(1)$$G_{xy}=\frac{4\rho JLf_{T}^{2}}{K}$$
where *L* and *r* are the sample length and density, respectively, and *f*_T_ is the fundamental frequency of the torsional vibration mode. *J* and *K* are given as follows:(2)$$J=\frac{BH\left(B^{2}+H^{2}\right)}{12}$$
and
(3)$$K=\frac{BH^{3}}{3}\left[1-\frac{192H}{\pi^{5}B}\sqrt{\frac{G_{xy}}{G_{xz}}}\cdot k\right]$$
where *B* is the sample width, *H* is the sample depth, *G_xz_* is the out-of-plane shear modulus, and *k* is derived as follows:(4)$$k=\overset{\infty }{\underset{n=1}{\sum }}\frac{\tanh\frac{\left(2n-1\right)\pi B}{2H}\sqrt{\frac{G_{xz}}{G_{xy}}}}{{\left(2n-1\right)}^{5}}$$

The calculation of the *G_xy_* value by using Equations (1)–(4) is complicated by the infinite series *k* contained in the *K* value. Therefore, the *G_xy_* value was calculated by using the following two modified equations. Firstly, the second term in the brackets of Equation (3) was ignored; therefore, *K* was approximated to be *BH*^3^/3 and
(5)$$G_{xy}=\frac{12\rho JL^{2}f_{T}^{2}}{BH^{3}}$$

Secondly, when the $B/H\sqrt{G_{xz}/G_{xy}}$ value is sufficiently high, the infinite series *k* can be approximated to 1. Additionally, supposing that *G_xy_* ≈ *G_xz_*, the *G_xy_* value can be approximated as follows:(6)$$G_{xy}=\frac{12\rho JL^{2}f_{T}^{2}}{BH^{3}\left(1-\frac{192H}{\pi^{5}B}\right)}$$

Equations (5) and (6) are convenient in that the *G_xy_* value is determined by using a single sample. However, the accuracy is reduced because the effect of the *G_xz_* value is ignored. Therefore, in addition to these equations, both the *G_xy_* and *G_xz_* values were determined by using additional data obtained from samples of different widths. From Equations (1)–(4), the following relation is derived by supposing that *k* = 1:(7)$$\frac{12\rho JL^{2}f_{T}^{2}}{H^{4}}=\frac{B}{H}G_{xy}-\frac{192}{\pi^{5}}G_{xy}\sqrt{\frac{G_{xy}}{G_{xz}}}$$

By conducting multiple torsional vibration tests of the samples with various *B/H* values, the 12*rJLf*_T_^2^/*H*^4^–*B/H* relation can be regressed into a linear relation with the slope and intercept of *a* and -*b*, respectively, which are represented as follows:(8)$$\left\{\begin{array}{c} \alpha =G_{xy}\;\;\;\;\;\;\;\;\;\;\;\;\;\;\;\;\\ \beta =\frac{192}{\pi^{5}}G_{xy}\sqrt{\frac{G_{xy}}{G_{xz}}} \end{array}\right.$$

Therefore,
(9)$$\frac{192}{\pi^{5}}\sqrt{\frac{G_{xy}}{G_{xz}}}=\frac{\beta }{\alpha }$$

By substituting the *b*/*a* value obtained as Equation (9) into Equations (1)–(4), the *G_xy_* value for each sample is calculated as follows:(10)$$G_{xy}=\frac{12\rho JL^{2}f_{T}^{2}}{BH^{3}\left(1-\frac{H}{B}\cdot \frac{\beta }{\alpha }\right)}$$

In the L-type sample, the *x* and *y* directions coincided with the L and T axes, respectively, and vice versa in the T-type sample. Therefore, the *G_xy_* values of the L- and T-type samples were denoted as *G*_LT_ and *G*_TL_, respectively. The *G*_LT_ and *G*_TL_ values calculated using Equations (5), (6), and (10) were compared with each other, and the accuracy of these equations were examined.

By using the *G_xy_* value obtained from each sample, as well as the *b* value obtained from the linear regression, the *G_xz_* value was obtained from Equation (8) as follows:(11)$$G_{xz}=G_{xy}{\left(\frac{192}{\pi^{5}}\cdot \frac{G_{xy}}{\beta }\right)}^{2}$$

The *G_xz_* value of the L- and T-type samples were denoted as *G*_LZ_ and *G*_TZ_, respectively.

By comparing the *G*_LT_, *G*_TL_, *G*_LZ_, and *G*_TZ_ values obtained from the different *B/H* values, the accuracy of the in-plane and out-of-plane shear moduli obtained from Equations (5), (6), (10), and (11) was investigated.

### 2.3. Flexural Vibration Tests

In addition to the torsional vibration tests, flexural vibration tests were conducted to measure the in-plane and out-of-plane shear moduli.

Figure 2a,b show the setup and diagram of the flexural vibration test, respectively. The initial length of the sample was 300 mm, whereas the initial width was 25 mm. The samples whose length direction coincided with the L and T axes were denoted as L- and T-type samples, respectively, and five samples were used per testing condition. After performing the flexural vibration tests, the sample length was decreased, and subsequent tests were consecutively performed. The length of the sample was shortened from 300 to 100 mm in 50 mm intervals. The sample was supported at the nodal positions of free-ends flexural vibration mode using prismatic wood blocks fixed on a plywood plate. The flexural vibration was generated by hitting the mid-width point at an end of the sample, which corresponds an anti-nodal position. The aerial vibration was detected and analyzed by using the same equipment used in the torsional vibration test. The resonance frequencies from the 1st to 4th flexural vibration modes were used for the analysis. The Young’s modulus in the length direction *E_x_* and the shear modulus in the length-depth plane, *G_xy_*, were calculated using the *X*–*Y* relation developed as the Timoshenko–Goens–Hearmon (TGH) method [32] represented as follows:(12)$$\left\{\begin{array}{c} X=\frac{48\pi^{2}\rho L^{2}f_{n}^{2}}{m_{n}^{4}H^{2}}\left[-2m_{n}F\left(m_{n}\right)+m_{n}^{2}F^{2}\left(m_{n}\right)\right]\;\\ Y=\frac{48\pi^{2}\rho L^{2}f_{n}^{2}}{m_{n}^{4}H^{2}}\left[\frac{H^{2}}{12L^{2}}+6m_{n}F\left(m_{n}\right)+m_{n}^{2}F^{2}\left(m_{n}\right)-\frac{4\pi^{2}\rho L^{2}f_{n}^{2}s}{G_{xy}}\right] \end{array}\right.$$
where *f*_n_ is resonance frequency of n^th^ flexural vibration mode; *s* is the Timoshenko’s shear factor, which is 1.2 for a sample with a rectangular cross section; and *m*_n_ and *F*(*m*_n_) are the coefficients corresponding to each resonance mode and are defined as follows:(13)$$\left\{\begin{array}{c} m_{1}=4.730\\ m_{2}=7.853\\ m_{n}=\frac{\left(2n+1\right)\pi }{2}\;\;\;\;\left(n\geq 3\right) \end{array}\right.$$
and
(14)$$\left\{\begin{array}{c} F\left(m_{1}\right)=0.9825\\ F\left(m_{2}\right)=1.0008\\ F\left(m_{n}\right)=1\;\;\;\;\left(n\geq 3\right) \end{array}\right.$$

The *E_x_* and *G_xy_* values were determined by regressing the *X*–*Y* relation into the following equation:(15)$$Y=E_{x}-s\frac{E_{x}}{G_{xy}}X$$

The refined *G_xy_* value was substituted into Equation (12), and the *E_x_* and *G_xy_* values were repeatedly calculated by using Equation (15) until they converged. The criterion of convergence was determined to be that the residual between the pre- and post-refined values of *E_x_/G_xy_* was smaller than 10^−6^.

When the *x* and *y* axes of the sample coincided with the L/T and T/L directions, respectively, *E_x_* = *E*_L_ and *E_x_* = *E*_T_, respectively, whereas *G_xy_* = *G*_LT_ and *G_xy_* = *G*_TL_, respectively (coordinate system A in Figure 2). In contrast, when the *x* and *y* axes of the sample coincided with the L/T and Z directions, respectively, *E_x_* = *E*_L_ and *E_x_* = *E*_T_, respectively, whereas *G_xy_* = *G*_LZ_ and *G_xy_* = *G*_TZ_, respectively (coordinate system B in Figure 2).

## 3. Results and Discussion

### 3.1. Torsional Vibration Tests

#### 3.1.1. In-Plane Shear Modulus

Figure 3 illustrates the 12*rJL*^2^*f*_T_^2^/*H*^4^–*B/H* relation and *a* and *b* values obtained from the linear regression. As shown in this figure, the coefficient of determination (*R*^2^) is close to 1; therefore, the approximation by Equation (7) is sufficiently accurate.

Figure 4 illustrates the *G*_LT_–*B, G*_LT_–*B/H, G*_TL_–*B*, and *G*_TL_–*B/H* relations obtained from the torsional vibration tests. Additionally, Table 2 lists the analysis of variance (Tukey tests) results for the *G*_LT_ and *G*_TL_ values in the different *B* and *B/H* values. The *G*_LT_ and *G*_TL_ values obtained by using Equation (5) increase as the *B* and *B/H* values increase. However, these values are obviously lower than those calculated by using Equations (6) and (10). In a previous study, Equation (5) was applied to the torsion test by using a square-plate to measure the in-plane shear modulus alone [14]. However, the accuracy could not be expected without conducting the FE analyses to reducing the effect of sample width. The tendency demonstrated in Figure 5 also demonstrates the irrelevancy of using Equation (5) without the aid of FE analysis. The *G*_LT_ and *G*_TL_ value obtained from Equation (5) ranges from 73 to 91% and from 67 to 89% of those obtained from Equation (10). In contrast, the *G*_LT_ and *G*_TL_ values obtained from Equation (6) range from 102 to 107% and from 98 to 100% of those obtained from Equation (10). Therefore, the values of in-plane modulus calculated by using Equations (6) and (10) are close to each other, and the dependence on the *B* and *B/H* values is moderated as illustrated in Figure 5. These results indicate that Equations (6) and (10) are more effective in reducing the dependence on the sample width than Equation (5). However, when using Equation (6), the *G*_LT_ value at *B* = 50 mm is significantly higher than those of the samples with the other *B* and *B/H* values at the significance level of 0.05. This tendency was induced by the difference between the *G*_LT_ and *G*_LZ_ values.

Table 3 lists the unpaired *t*-test results performed on the difference in the *G*_LT_ and *G*_TL_ values calculated by using Equations (5), (6), and (10), as well as their corresponding *B* and *B/H* values. The statistical analysis also indicates that the *G*_LT_ and *G*_TL_ values calculated by using Equation (5) are significantly lower than those calculated by using Equations (6) and (10). Despite the closeness between the *G*_LT_–*B* and *G*_LT_–*B/H* relations shown in Figure 4, the *G*_LT_ values obtained from Equation (6) are 102–107% of those calculated by using Equation (10). Thus, they are significantly higher than that obtained from the Equation (10), and Equation (6) cannot effectively reduce the effect of the out-of-plane shear modulus. In contrast, the *G*_TL_ values obtained from Equation (6) are 98–100% of those calculated by using Equation (10), and there is no significant difference between the *G*_TL_ values obtained from Equations (5) and (9).

Table 4 lists the average values of *G*_LT_ and *G*_TL_. The *G*_LT_ and *G*_TL_ values should coincide with each other, and the values calculated by using Equation (10) satisfy this condition. Table 4 also indicates that the coefficients of variation of the *G*_LT_ and *G*_TL_ derived using Equation (5) are often close to 10%, whereas those derived using Equations (6) and (10) are less than 3% and 2%, respectively. As represented in Figure 5 and Table 2, the dependence of the *G*_LT_ and *G*_TL_ values on the *B* and *B/H* values is effectively reduced when using Equation (10). This phenomenon also indicates that the torsional vibration tests when using Equation (10) are more advantageous than the square-plate twist tests in which the correction by the FE calculation is required [18,30]. Additionally, the *G*_LT_ and *G*_TL_ values listed in Table 4 are often higher than those obtained from previous studies listed in Table 1, although the density of the samples (=29.4 ± 0.5 kg/m^3^) are often lower than the materials used in these studies. These higher values are due to the effective reduction of stress concentration, which is often induced in the static tests based on EN 12090 and ASTM C273/C273M-11 [14,19,20,21].

From these results, when using Equation (10), the variation in obtaining the in-plane shear modulus is effectively restricted. These results indicate that Equation (10) is promising for calculating the in-plane shear modulus with a high accuracy, using the data obtained from the torsional vibration tests alone, although multiple samples with different configurations are required.

#### 3.1.2. Out-of-Plane Shear Modulus

Figure 5 illustrates the *G*_LZ_–*B, G*_LZ_–*B/H, G*_TZ_–*B*, and *G*_TZ_–*B/H* relations obtained by using Equation (10). Similar to the *G*_LT_ and *G*_TL_ values calculated by using Equation (10), the dependence of the *G*_LZ_ and *G*_TZ_ values on the *B* and *B/H* values is not significant; therefore, Equation (11) is effective for determining the out-of-plane shear modulus with the reduced effect of sample configuration.

**Figure 5 polymers-14-01148-f005:**
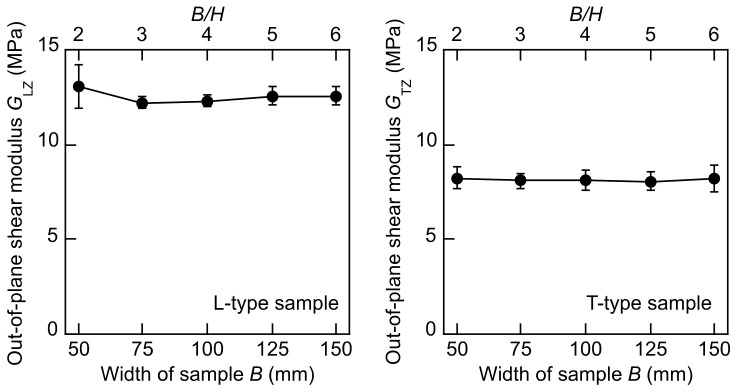
*G*_LZ_–*B, G*_LZ_–*B/H*, *G*_TZ_–*B*, and *G*_TZ_–*B/H* relations obtained from the torsional vibration tests. Results = average ± standard deviations.

Table 5 lists the *G*_LZ_ and *G*_TZ_ values of the total 25 samples. The *G*_LZ_ value is higher than the *G*_LT_ value listed in Table 4. Therefore, the high value of *G*_LZ_ affects the *G*_LT_ value when the *G*_LZ_ value is calculated by using Equation (6). In contrast, there is little difference between the *G*_TL_ and *G*_TZ_ values; therefore, the *G*_TZ_ value does not affect the measurement of the *G*_TL_ value. It was considered that the direction of extrusion dominated the cell structure in the XPS, and this affected the anisotropy in these shear moduli. Further research should be undertaken to examine the relation between the cell structure and anisotropy in more detail. Comparing the results listed in Table 4 and Table 5, we can see that the coefficient of variation of the out-of-plane shear modulus is larger than that of the in-plane shear modulus. In this study, the *B* value was always larger than the *H* value; therefore, the accuracy in measuring the out-of-plane shear modulus was not superior to that in measuring the in-plane shear modulus. However, as described below, the coefficients of variation obtained from the torsional vibration tests are obviously lower than those obtained from the flexural vibration tests. Therefore, the torsional vibration test is also recommended to measure the out-of-plane shear modulus.

### 3.2. Flexural Vibration Tests

#### 3.2.1. Young’s Modulus in the Length and Width Directions of the XPS Panel

Figure 6 illustrates the *E*_L_–*L*, *E*_L_–*L/H*, *E*_T_–*L*, and *E*_T_–*L/H* relations obtained from the flexural vibration tests. The *E*_L_ and *E*_T_ values could be obtained while reducing the effect of the *L* value.

Equations (12)–(15) suggest that the accuracy in measuring the Young’s modulus affects the shear modulus value in the flexural vibration tests. However, the standard deviations of the Young’s modulus are obviously larger than those of the shear modulus obtained from the torsional vibration test. Therefore, although the dependence of the *L* value on the *E*_L_ and *E*_T_ values are not significant, the *E*_L_ values obtained from the L-type samples are dependent of the depth direction of the sample, whereas such dependence is not found in the results obtained from the T-type samples. Further research should be undertaken to clarify the source of these tendencies, as this was not the principal aim of this study.

#### 3.2.2. In-Plane and Out-of-Plane Shear Moduli

The *G*_LT_–*L, G*_LT_–*L/H, G*_TL_–*L*, and *G*_TL_–*L/H* relations and *G*_LZ_–*L*, *G*_LZ_–*L/H, G*_TZ_–*L*, and *G*_TZ_–*L/H* relations are demonstrated in Figure 7 and Figure 8, respectively. Similar to the torsional vibration test, Tukey tests were conducted for the shear moduli corresponding to the different *L* values, and it was revealed that the dependence of the shear moduli on the *L* value was not significant. Therefore, the TGH method is effective for obtaining the shear modulus while reducing the effect of the sample length.

Table 6 lists the *G*_LT_, *G*_TL_, *G*_LZ_, and *G*_TZ_ values. Commonly, can be the results obtained from the torsional vibration tests listed in Table 4 and Table 5, the *G*_LZ_ value is higher than the *G*_LT_, *G*_TL_, and *G*_LZ_ values. However, the coefficients of variation of the shear moduli obtained from the flexural vibration tests are obviously higher than those obtained from the torsional vibration tests. In the flexural vibration tests, sample configuration is extremely dominant in terms of the accuracy in determining values of both the Young’s modulus and shear modulus. As the Young’s modulus-to-shear modulus ratio is small, the length-to-depth ratio should be sufficiently reduced when measuring these moduli [33]. Therefore, there is a concern that the length-to-depth ratio of the sample was not small enough to guarantee the accuracy of the Young’s modulus-to-shear modulus ratio, and due to the inaccuracy, the variation in the shear modulus was pronounced. The torsional vibration test was free from such an irrelevancy; therefore, the shear modulus values could be accurately obtained while restricting the variation.

Summarizing the abovementioned results, the multiple torsional vibration tests are promising for obtaining both the in-plane and out-of-plane shear moduli of XPS panel with a high accuracy when the widths of the samples are varied.

## 4. Conclusions

Torsional vibration tests of extruded polystyrene (XPS) panel were conducted to measure both the in-plane and out-of-plane shear moduli by varying the widths of the sample. The accuracy of the shear moduli was evaluated from the statistical analyses (Tukey tests and unpaired *t*-tests) and comparison with the shear modulus values obtained from flexural vibration tests conducted independently of the torsional vibration tests.

The out-of-shear modulus in the length-thickness plane was significantly higher than the in-plane shear modulus; therefore, the anisotropy in these shear moduli was significant. The in-plane shear modulus that was determined by using a single sample was often dependent on the sample width because of the anisotropy, and the coefficient of variation of the in-plane shear modulus was 7–10%. However, when using the multiple data obtained by varying the sample widths, the abovementioned dependence was relevantly reduced. Additionally, the variation in shear modulus was restricted below 2% and 7% for the in-plane and out-of-plane shear moduli, respectively, which were obviously lower than those obtained from the flexural vibration tests (approximately 20%). In the simple shear test frequently conducted in several foam materials, stress concentration at the gripped portion becomes an obstacle for determining the shear modulus accurately. Additionally, in the flexural vibration tests, the sample configuration was extremely dominant when determining the accuracy in measuring the shear modulus. The torsional vibration tests using a range of samples of various widths were free from these drawbacks; therefore, this method was recommended to measure both the in-plane and out-of-plane shear moduli of XPS with a high accuracy.

## Figures and Tables

**Figure 1 polymers-14-01148-f001:**
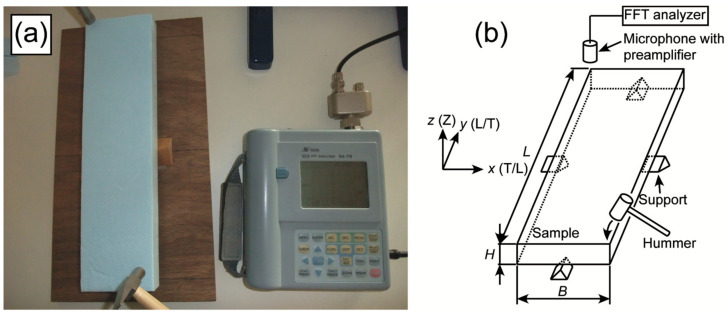
Setup (**a**) and diagram (**b**) of the torsional vibration test.

**Figure 2 polymers-14-01148-f002:**
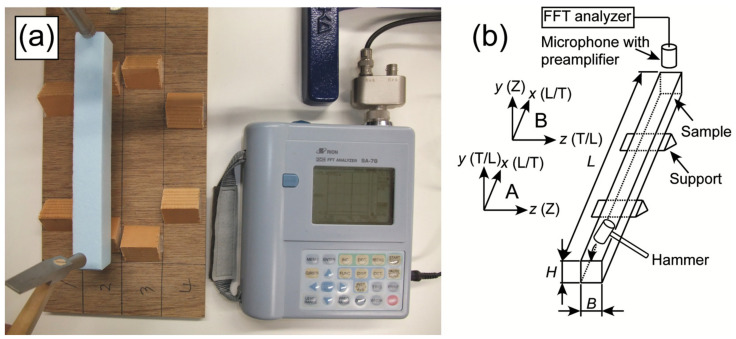
Setup (**a**) and diagram (**b**) of the flexural vibration test. A and B in (**b**) represent the coordinate system used for measuring the *G*_LT_ and *G*_TL_ values and the *G*_LZ_ and *G*_TZ_ values, respectively.

**Figure 3 polymers-14-01148-f003:**
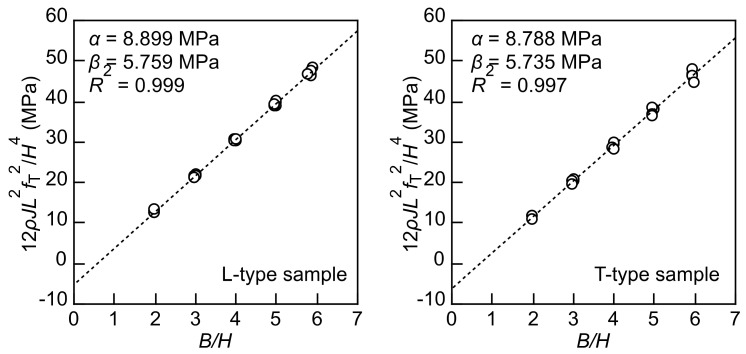
The 12*rJLf*_T_^2^/*H*^4^–*B/H* relation and *a* and *b* values obtained from the linear regression represented by the dashed line.

**Figure 4 polymers-14-01148-f004:**
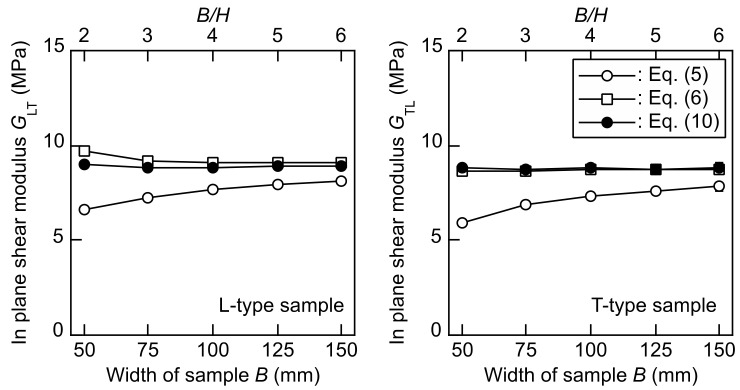
*G*_LT_–*B, G*_LT_–*B/H, G*_TL_–*B*, and *G*_TL_–*B/H* relations obtained from the torsional vibration tests. Results = average ± standard deviations.

**Figure 6 polymers-14-01148-f006:**
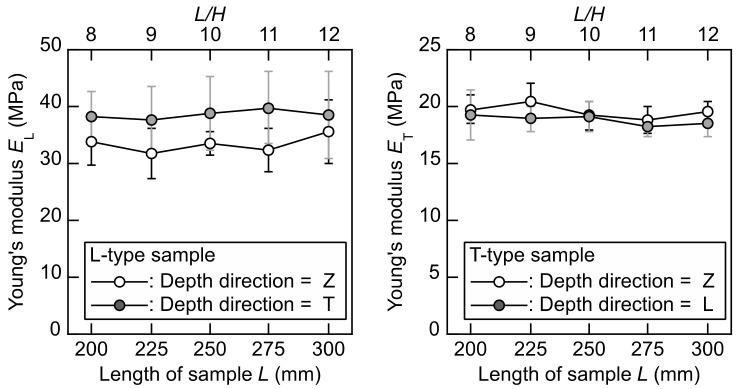
*E*_L_–*L*, *E*_L_–*L/H*, *E*_T_–*L*, and *E*_T_–*L/H* relations determined by flexural vibration tests. Results = average ± standard deviations.

**Figure 7 polymers-14-01148-f007:**
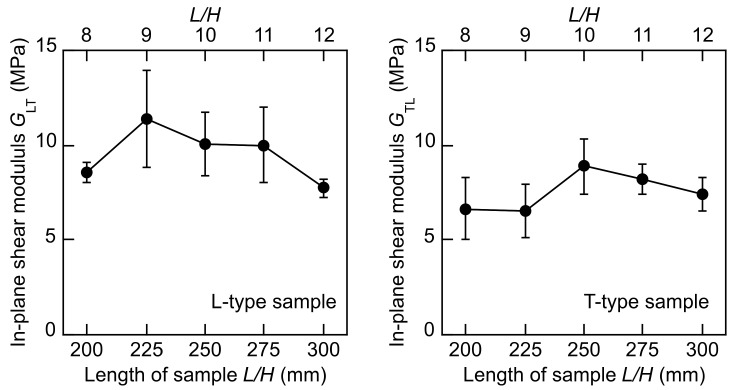
*G*_LT_–*L, G*_LT_–*L/H, G*_TL_–*L*, and *G*_TL_–*L/H* relations determined by flexural vibration tests. Results = average ± standard deviations.

**Figure 8 polymers-14-01148-f008:**
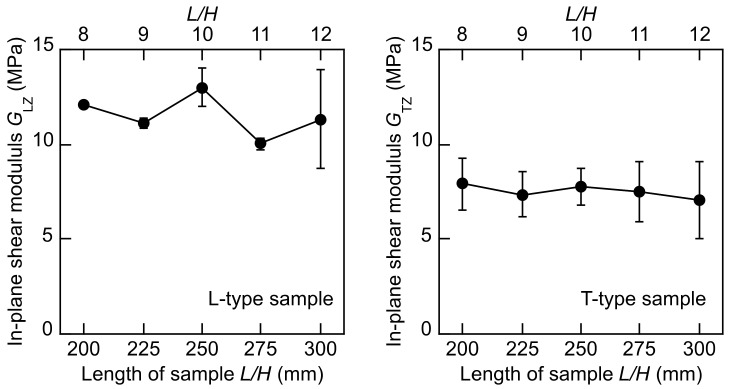
*G*_LZ_–*L, G*_LZ_–*L/H, G*_TZ_–*L*, and *G*_TZ_–*L/H* relations determined by flexural vibration tests. Results = average ± standard deviations.

**Table 1 polymers-14-01148-t001:** Shear modulus of XPS and EPS obtained by various methods referred from previous studies.

Reference	Material	Test Method	Density (kg/m^3^)	Shear Modulus (MPa)
Static method				
Kilar et al. [14]	XPS	EN 12090	45	7.46
Yoshihara et al. [18]	XPS	ASTM C273	32	6.55
Yoshihara and Maruta [18]	XPS	ISO 15310	32	5–9
Yoshihara and Maruta [23]	XPS	AFPB	32	10.4
Taskin et al. [19]	XPS	EN 12090	23.40	4.32
Gnip et al. [20]	EPS	EN 12090	10–35	1–5
Vėjelis et al. [21]	EPS	EN 12090	11–30	0.9–4.5
Dynamic method				
Yoshihara et al. [18]	XPS	FV	32	8–10
Athanasopoulous et al. [27]	EPS	RC	17.1	4.9
Ossa [28]	EPS	RC	24	10
Abd-El-Mottaleb et al. [29]	EPS	US	25–35	1.5–2.7
Yoshihara and Maruta [30]	XPS	TVSP	32	6–8

AFPB, FV, TVSP, RC, and US are the abbreviation of asymmetric four-point bending, flexural vibration, torsional vibration of square-plate, resonant column, and ultrasonic methods, respectively.

**Table 2 polymers-14-01148-t002:** Tukey test results for the *G*_LT_ and *G*_TL_ values corresponding to the different *B* and *B/H* values.

		*G* _LT_			*G* _TL_		
*B* (mm)	*B/H*	Equation (5)	Equation (6)	Equation (10)	Equation (5)	Equation (6)	Equation (10)
50	2			A		A	A
75	3		A	A		A	A
100	4		A	A	AB	A	A
125	5	A	A	A	AB	A	A
150	6	A	A	A	B	A	A

Significance level is greater than 0.05 among the samples with the same letters in a same column.

**Table 3 polymers-14-01148-t003:** Results of the unpaired *t*-test results performed on the difference in the *G*_LT_ and *G*_TL_ values calculated by using Equations (5), (6), and (10) corresponding to the *B* and *B/H* values.

		*G* _LT_			*G* _TL_		
*B* (mm)	*B/H*	Equations (5) and (6)	Equations (5) and (10)	Equations (6) and (10)	Equations (5) and (6)	Equations (5) and (10)	Equations (6) and (10)
50	2	SA	SA	SA	SA	SA	NS
75	3	SA	SA	SA	SA	SA	NS
100	4	SA	SA	SA	SA	SA	NS
125	5	SA	SA	SB	SA	SA	NS
150	6	SA	SA	SB	SA	SA	NS

SA and SB indicate that the significance levels are lower than 0.01 and 0.05, respectively, whereas NS indicates that the significance level is greater than 0.05.

**Table 4 polymers-14-01148-t004:** *G*_LT_ and *G*_TL_ values calculated using Equations (5), (6), and (10).

*G*_LT_ (MPa)			*G*_TL_ (MPa)		
Equation (5)	Equation (6)	Equation (10)	Equation (5)	Equation (6)	Equation (10)
7.52 ± 0.56	9.24 ± 0.27	8.91 ± 0.15	7.11 ± 0.71	8.71 ± 0.18	8.87 ± 0.09
(7.45)	(2.92)	(1.68)	(9.99)	(2.07)	(1.01)

Data = average ± standard deviations. Value in the parentheses represent the coefficient of variation (%).

**Table 5 polymers-14-01148-t005:** *G*_LZ_ and *G*_TZ_ values calculated by using Equation (11).

*G*_LZ_ (MPa)	*G*_TZ_ (MPa)
12.52 ± 0.64 (5.11)	8.14 ± 0.51 (6.27)

Data = average ± standard deviations. Values in the parentheses represent the coefficient of variation (%).

**Table 6 polymers-14-01148-t006:** *G*_LT_, *G*_TL_, *G*_LZ_, and *G*_TZ_ values obtained from flexural vibration tests.

*G*_LT_ (MPa)	*G*_TL_ (MPa)	*G*_LZ_ (MPa)	*G*_TZ_ (MPa)
9.56 ± 2.00	7.54 ± 1.52	11.52 ± 1.93	7.52 ± 1.39
(20.9)	(20.2)	(16.8)	(18.5)

Data = average ± standard deviations. Values in the parentheses represent the coefficient of variation (%).

## Data Availability

Data is contained within the article.

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
