# Peer review of "Torsional Vibration Tests of Extruded Polystyrene with Improved Accuracy in Determining the Shear Modulus"

_polymers, 2022, doi:10.3390/polym14061148_

Round 1

Reviewer 1 Report

  1. At this moment the paper is only a sum of results using a few relations and some experimental tests.
  2. It is not clear why the authors select the parameters used in their experiments (see, for instance, parameter B in Table 1, 2). What happens choosing other values?
  3. There is no justification concerning the convergence of their algorithm (see page 5 where the authors state that the new values of E_x and G_xy are repeatedly calculated until they converge.
  4. Conclusions have to be improved. The main conclusion is written in a single statement at the end of the manuscript.

Author Response

To Reviewer #1

Thank you very much for your suggestions and advice. We revised the manuscript and provided a point-by-point to your comments. Followings are the response to your comments:

  1. At this moment the paper is only a sum of results using a few relations and some experimental tests.

Answer:

Considering the comments from you and another reviewer, the manuscript is revised with addressing the results obtained in previous studies and advantage of our method. If you still think that the revisions are not still adequate, please inform the issue. We would like to re-revise it again.

  1. It is not clear why the authors select the parameters used in their experiments (see, for instance, parameter B in Table 1, 2). What happens choosing other values?

Answer:

In the torsion test of anisotropic material, the shear modulus value is dependent on the aspect ratio B/H. This issue also indicates that both the in-plane and out-of-plane shear moduli can be measured when varying the B/H values. Considering this issue, the torsional vibration tests were conducted under various B/H value. In the test, the thickness of the sample H is constant (= 25 mm); therefore, the B value was selected as the parameter to vary the B/H value. However, to pronounce the issue that the B/H is the principal parameter to determine the in-plane and out-of-plane shear moduli, following descriptions are addressed in the lines 223-236:

In the torsion test of anisotropic material, the effects of the in-plane and out-of-plane shear moduli vary according to the aspect ratio B/H [31]. Therefore, it is expected to obtain both the shear moduli can be measured when using the data obtained from multiple torsion tests with varying the B/H value.

Additionally, Figs. 5 and 6 and Tables 2 and 3 are revised with considering the B/H values. Moreover, in the flexural vibration tests, the length/depth ratio L/H is the principal parameter to determine the Young’s modulus and shear modulus; therefore, Figs. 7-9 are revised as those including the L/H values.

  1. There is no justification concerning the convergence of their algorithm (see page 5 where the authors state that the new values of E_x and G_xy are repeatedly calculated until they converge.

Answer:

Considering your suggestion, the criterion of convergence was denoted in the revised version in the lines 315-317 as:

The criterion of convergence was determined as that the residual between the pre- and post-refined values of Ex/Gxy was smaller than 10-6.

We think that both the Ex and Gxy values are converged enough under this criterion.

4.Conclusions have to be improved. The main conclusion is written in a single statement at the end of the manuscript.

Answer:

Thank you very much for your suggestion. Considering your suggestion, following conclusion is addressed in the revised version.

Torsional vibration tests of extruded polystyrene (XPS) panel were conducted to measure both the in-plane and out-of-plane shear moduli by varying the widths of the sample. The accuracy of the shear moduli was evaluated from the statistical analyses (Tukey tests and unpaired t-tests) and comparison with the shear modulus values obtained from flexural vibration tests conducted independently of the torsional vibration tests.

The out-of-shear modulus in the length-thickness plane was significantly higher than the in-plane shear modulus; therefore, the anisotropy in these shear moduli was significant. The in-plane shear modulus determined using a single sample was often dependent on the sample width because of the anisotropy, and the coefficient of variation of the in-plane shear modulus was often close to 10%. However, when using the multiple data obtained by varying the sample widths, the abovementioned dependence was relevantly reduced. Additionally, the variation in shear modulus was restricted below 2% and 7% for the in-plane and out-of-plane shear moduli, respectively, which were obviously lower than those obtained from the flexural vibration tests (approximately 20%). In the simple shear test frequently conducted in several foam materials, stress concentration at the gripped portion becomes an obstacle for determining the shear modulus accurately. Additionally, in the flexural vibration tests, the sample configuration extremely dominant the accuracy in measuring the shear modulus. The torsional vibration tests using a range of samples of various widths were free from these drawbacks; therefore, this method was recommended to measure both the in-plane and out-of-plane shear moduli of XPS with a high accuracy.

Thank you very much again for your fruitful suggestions and advice. I hope that the revised manuscript is now acceptable for publication. When you think that there are still any insufficient descriptions, however, please inform the issue. I’d like to consider the re-revision as soon as possible.

Yours sincerely,

Hiroshi YOSHIHARA

Faculty of Science and Engineering

Shimane University

Matsue, Shimane

Shimane 690-8504, Japan

Phone: +81-852-32-6508

Reviewer 2 Report

Thank you for submitting your paper. The work done here draws attention to a significant subject in performance of extruded polystyrene. I have found the paper to be interesting. However, several issues need to be addressed properly before the paper is being considered for publication. My comments including major and minor concerns are given below:

  1. Please consider reviewing the abstract and highlight the novelty, major findings, and conclusions. I suggest reorganizing the abstract, highlighting the novelties introduced. The abstract should contain answers to the following questions:
  2. What problem was studied and why is it important?
  3. What methods were used?
  4. What conclusions can be drawn from the results? (Please provide specific results and not generic ones).
  5. The abstract must be improved. It should be expanded. Please use numbers or % terms to clearly shows us the results in your experimental work. Please expand the abstract.
  6. Please consider reporting on studies related to your work from mdpi journals.
  7. The introduction must be expanded, please consider improving the introduction, provide more in-depth critical review about past studies similar to your work, mention what they did and what were their main findings then highlight how does your current study brings new difference to the field.
  8. In the materials and methods section, the author should add some images for the tests setup, fabricated samples, equipment used and so on. After all, this is an experimental study and such details are necessary to provide clear details about the work done here.
  9. The authors should add a list of nomenclature at the end of the manuscript which contains all symbols, Greek letters, abbreviations ..etc.
  10. Standards used for the torsional/flexural vibration tests should be indicated.
  11. Line 101, how much does excluding the Gxz affect the accuracy, 
  12. Line 171 “are close to each other” please use % or numbers to clearly show the readers how close they are to each other.
  13. In Figure 5 how can the authors correlate the effect of sample with with the corresponding accuracy of the results?
  14. Lines 189-190 does not read well, please rephrase.
  15. Conclusion is missing?
  16. The results are merely described and is limited to comparing the experimental observation and describing results. The authors are encouraged to include a more detailed results and discussion section and critically discuss the observations from this investigation with existing literature.

Author Response

To Reviewer #2

Thank you very much for your suggestions and advice. We revised the manuscript and provided a point-by-point to your comments. Followings are the response to your comments:

  1. Please consider reviewing the abstract and highlight the novelty, major findings, and conclusions. I suggest reorganizing the abstract, highlighting the novelties introduced. The abstract should contain answers to the following questions:

Answer:

Thank you for your comments on the abstract. The abstract is revised while considering your suggestions as follows:

Recently, extruded polystyrene (XPS) foam has been often used as a component of construction materials; therefore, it is important to characterize its mechanical properties, including shear modulus. Despite the importance, it is often difficult to determine the shear modulus accurately by many conventional methods; therefore, it is desirable to establish another method to measure the shear modulus with a high accuracy. Among various methods, torsional vibration test is advantageous because it can be performed easily under the pure shear stress condition in the test sample and both the in-plane and out-of-plane shear moduli can be obtained. However, it is difficult to find any examples performing the torsional vibration tests. In this study, the in-plane and out-of-plane shear moduli of XPS were determined through torsional vibration tests using samples of various widths. In addition, the shear moduli were also determined through flexural vibration tests and compared with those obtained from the torsional vibration tests. In the torsional vibration tests, the anisotropy in these shear moduli became an obstacle, and the in-plane shear modulus determined using a single sample was often dependent on the width/thickness ratio of the sample. In this condition, the coefficient of variation of the in-plane shear modulus value was often close to 10%. However, when using data obtained from the samples with various width/thickness ratios, both the in-plane and out-of-plane shear moduli could be obtained while reducing the abovementioned dependence. Additionally, the coefficients of variation were restricted to those below 2% and 7% for the in-plane and out-of-plane shear moduli, respectively, and these values were obviously lower than those obtained from the flexural vibration tests (approximately 20%). In the proposed method, both the in-plane and out-of-plane shear moduli can be obtained accurately without using any numerical analyses, which are often required in the standardized methods to improve the accuracy. Thus, for accurate measurement of both types of shear moduli of XPS, we recommend performing torsional vibration tests using a range of samples of various width/thickness ratios.

  1. What problem was studied and why is it important?

Answer:

In the present status, it is difficult to obtain the shear modulus of XPS with a high accuracy, despite the increase of the necessity of XPS as construction materials. Therefore, the establishment of the method for measuring the shear modulus is an essential issue for the effective use of XPS in construction. This issue is described in the abstract. Additionally, we think that the measurement method should be evolved to a standardized method in a future. In addition to the abstract, the descriptions are addressed in the lines 191-203 as follows:

Considering the simplicity and practicality, it is desirable to determine the shear modulus accurately from the testing data alone without performing any FE calculations. Additionally, the method for measuring the shear modulus of XPS has not been determined in major standards, including ASTM, EN, ISO, and JIS. When considering the extension of the use of XPS as construction materials, it is necessary to establish the measurement method of the shear modulus for future standardization. To measure the shear modulus, torsional vibration test is more advantageous than other methods because it can be performed easily under the pure shear stress condition in the test sample. Therefore, when the effect of sample configuration can be reduced, the shear modulus can be measured with a high accuracy, and it is promising that the torsional vibration test becomes a candidate for measuring the shear modulus of XPS in the major standards. However, it is difficult to find the examples applying the torsional vibration tests to measure the shear modulus of XPS.

  1. What methods were used?

Answer:

We describe that the in-plane and out-of-plane shear moduli are determined from the torsional vibration and flexural vibration tests in the abstract. However, we have a concern that we misunderstand the meaning of your comment. If the response is not irrelevant, please inform the issue.

  1. What conclusions can be drawn from the results? (Please provide specific results and not generic ones).

Answer:

The conclusion is addressed with the specific % terms (coefficient of variation) in the abstract, so we think that the specific results can be demonstrated in the abstract.

Additionally, the conclusion is addressed as Section 4, so please see below or the descriptions in the revised version. We believe that the specific results are demonstrated also in the conclusion. However, if the descriptions are not still adequate, please inform the issue.

  1. The abstract must be improved. It should be expanded. Please use numbers or % terms to clearly shows us the results in your experimental work. Please expand the abstract.

Answer:

The coefficient of variation is used to represent the consistency of the shear modulus obtained using the samples with various widths. As in the reply to Comment 4, the “% terms” for the coefficient of variation is represented in the abstract.

  1. Please consider reporting on studies related to your work from mdpi journals.

Answer:

Considering your advice, the papers on the thermal insulation performance of XPS published in several mdpi journals are listed as Refs. 1, 3, 4, and 5. Unfortunately, it is difficult to find relevant papers on the shear properties of soft plastics including XPS in the mdpi journals except for Ref. 23, which is our paper published in Polymers.

  1. The introduction must be expanded, please consider improving the introduction, provide more in-depth critical review about past studies similar to your work, mention what they did and what were their main findings then highlight how does your current study brings new difference to the field.

Answer:

Thank you for your fruitful suggestion. Considering your advice, the introduction section is expanded while quoting the previous studies (with the data of density and shear modulus in Table 1) and problems of the methods in these studies. Additionally, the highlight of the study conducted here is described in the lines 193-205 as:

Considering the practical aspect, it is desirable to determine the shear modulus accurately from the testing data alone without performing the FE calculations. Additionally, the method for measuring the shear modulus of XPS has not been determined in major standards, including ASTM, EN, ISO, and JIS. When considering the extension of the use of XPS as construction materials, it is necessary to establish the measurement method of the shear modulus for future standardization. To measure the shear modulus, torsional vibration test is more advantageous than other methods because it can be performed easily under the pure shear stress condition in the test sample. Therefore, when the effect of sample configuration can be reduced, the shear modulus can be measured with a high accuracy, and it is promising that the torsional vibration test becomes a candidate for measuring the shear modulus of XPS in the major standards. However, it is difficult to find the examples applying the torsional vibration tests to measure the shear modulus of XPS.

Please see the revised version, and if you think that the revisions are not adequate yet, please inform the issue.

  1. In the materials and methods section, the author should add some images for the tests setup, fabricated samples, equipment used and so on. After all, this is an experimental study and such details are necessary to provide clear details about the work done here.

Answer:

Thank you for your suggestion. According to your suggestion, photographs of the setup are shown as well as the diagrams of the tests in Figs. 1 and 2. The sample and apparatus are included in the photographs, and the details are demonstrated in the diagrams. We think that the presentation of these figures is improved over the original version.

  1. The authors should add a list of nomenclature at the end of the manuscript which contains all symbols, Greek letters, abbreviations ..etc.

Answer:

Considering your advice, nomenclature section is addressed below the conclusion. About the format of the section, we would like to follow the instruction from the managing editor.

  1. Standards used for the torsional/flexural vibration tests should be indicated.

Answer:

Unfortunately, torsional/flexural vibration methods have not been determined in major standards. However, we think that the absence of the standards is exactly the opportunity for the standardization of torsional vibration test as the method for measuring the shear modulus of XPS. This issue is addressed in the lines 191-203 as follows:

Considering the simplicity and practicality, it is desirable to determine the shear modulus accurately from the testing data alone without performing any FE calculations. Additionally, the method for measuring the shear modulus of XPS has not been determined in major standards, including ASTM, EN, ISO, and JIS. When considering the extension of the use of XPS as construction materials, it is necessary to establish the measurement method of the shear modulus for future standardization. To measure the shear modulus, torsional vibration test is more advantageous than other methods because it can be performed easily under the pure shear stress condition in the test sample. Therefore, when the effect of sample configuration can be reduced, the shear modulus can be measured with a high accuracy, and it is promising that the torsional vibration test becomes a candidate for measuring the shear modulus of XPS in the major standards. However, it is difficult to find the examples applying the torsional vibration tests to measure the shear modulus of XPS.

  1. Line 101, how much does excluding the Gxz affect the accuracy,

Answer:

Considering your suggestion, the effect in excluding the Gxz value is numerically represented. The corresponding descriptions (lines 376-382) are revised as follows:

Despite the closeness between the GLT-B and GLT-B/H relations shown in Figure 4, the GLT values obtained from Eq. (6) are 102-107% of those calculated using Eq. (10). Thus, they are significantly higher than that obtained from the Eq. (10), and Eq. (6) cannot effectively reduce the effect of the out-of-plane shear modulus. In contrast, the GTL values obtained from Eq. (6) are 98-100% of those calculated using Eq. (10), and there is not significant in the difference between the GTL values obtained from Eqs. (5) and (9).

  1. Line 171 “are close to each other” please use % or numbers to clearly show the readers how close they are to each other.

Answer:

Similar to the response to Comment 11, the closeness is numerically represented. The corresponding descriptions (lines 348-363) are revised as follows:

The GLT and GTL value obtained from Eq. (5) ranges from 73 to 91% and from 67 to 89% of those obtained from Eq. (10). In contrast, the GLT and GTL value obtained from Eq. (6) ranges from 102 to 107% and from 98 to 100% of those obtained from Eq. (10). Therefore, the values of in-plane modulus calculated using Eqs. (6) and (10) are close to each other, and the dependence on the B and B/H values is moderated as illustrated in Figure 5. These results indicate that Eqs. (6) and (10) are more effective in reducing the dependence on the sample width than Eq. (5).

  1. In Figure 5 how can the authors correlate the effect of sample width with the corresponding accuracy of the results?

Answer:

We think that Eq. (10) effectively reduces the dependence of the GLT and GTL values on the sample width as the results shown in Figure 5 and Table 2, and the reduction of the dependence consequently leads the coincidence of the GLT and GTL values and small values of the standard deviations. This issue is described in the lines 424-428 as follows:

Table 4 also indicates that the coefficients of variation of the GLT and GTL derived using Eq. (5) are close to 10%, whereas those derived using Eqs. (6) and (10) are less than 3% and 2%, respectively. As represented in Figure 5 and Table 2, the dependence of the GLT and GTL values on the B and B/H values is effectively reduced when using Eq. (10).

If the descriptions are still inadequate, please inform the issue. We would like to re-revise them again.

  1. Lines 189-190 does not read well, please rephrase.

Answer:

The issue that the effect of the out-of-plane shear modulus on the measurement of the in-plane shear modulus is moderated when both the shear modulus values are close with each other. Therefore, to reduce the misunderstandings of the readers, the sentence “because the GTL and GTZ values are close to each other” is removed from the manuscript.

  1. Conclusion is missing?

Answer:

Thank you very much for your suggestion. Commonly to the reply to Comment 15, following conclusion is addressed in the revised version.

Torsional vibration tests of extruded polystyrene (XPS) panel were conducted to measure both the in-plane and out-of-plane shear moduli by varying the widths of the sample. The accuracy of the shear moduli was evaluated from the statistical analyses (Tukey tests and unpaired t-tests) and comparison with the shear modulus values obtained from flexural vibration tests conducted independently of the torsional vibration tests.

The out-of-shear modulus in the length-thickness plane was significantly higher than the in-plane shear modulus; therefore, the anisotropy in these shear moduli was significant. The in-plane shear modulus determined using a single sample was often dependent on the sample width because of the anisotropy, and the coefficient of variation of the in-plane shear modulus was often close to 10%. However, when using the multiple data obtained by varying the sample widths, the abovementioned dependence was relevantly reduced. Additionally, the variation in shear modulus was restricted below 2% and 7% for the in-plane and out-of-plane shear moduli, respectively, which were obviously lower than those obtained from the flexural vibration tests (approximately 20%). In the simple shear test frequently conducted in several foam materials, stress concentration at the gripped portion becomes an obstacle for determining the shear modulus accurately. Additionally, in the flexural vibration tests, the sample configuration extremely dominant the accuracy in measuring the shear modulus. The torsional vibration tests using a range of samples of various widths were free from these drawbacks; therefore, this method was recommended to measure both the in-plane and out-of-plane shear moduli of XPS with a high accuracy.

  1. The results are merely described and is limited to comparing the experimental observation and describing results. The authors are encouraged to include a more detailed results and discussion section and critically discuss the observations from this investigation with existing literature.

Answer:

Thank you very much for your suggestion. The comparison between the present and past studies are addressed in the lines 423-435 as:

Table 4 lists the average values of GLT and GTL. The GLT and GTL values should coincide with each other, and the values calculated using Eq. (10) satisfy this condition. Table 4 also indicates that the coefficients of variation of the GLT and GTL derived using Eq. (5) are close to 10%, whereas those derived using Eqs. (6) and (10) are less than 3% and 2%, respectively. As represented in Figure 5 and Table 2, the dependence of the GLT and GTL values on the B and B/H values is effectively reduced when using Eq. (10). This phenomenon also indicates that the torsional vibration tests with using Eq. (10) are more advantageous than the square-plate twist tests in which the correction by the FE calculation is required [18, 30]. Additionally, the GLT and GTL values listed in Table 4 are often higher than those obtained from previous studies listed in Table 1, although the density of the samples (= 29.4 ± 0.5 kg/m3) are often lower than the materials used in these studies. These higher values are due to the effective reduction of stress concentration, which is often induced in the static tests based on EN 12090 and ASTM C273/C273M-11 [14, 19-21].

Thank you very much for your fruitful suggestions and advices. I hope that the revised manuscript is now acceptable for publication. When you think that there are still any insufficient descriptions, however, please inform the issue. I’d like to consider the re-revision as soon as possible.

Yours sincerely,

Hiroshi YOSHIHARA

Faculty of Science and Engineering

Shimane University

Matsue, Shimane

Shimane 690-8504, Japan

Phone: +81-852-32-6508

Round 2

Reviewer 1 Report

The authors performed the requirements mentioned in my previous review. From my point of view, the manuscript fulfils now the criteria for the publication.

Reviewer 2 Report

The authors provided the answers to the comments from the first round of review and made sufficient changes in the manuscript according to these comments. I recommend this manuscript for a publication in its present form.